

# First year of the new Arctic AWIPEV-COSYNA cabled Underwater Observatory in Kongsfjorden, Spitsbergen.

Philipp Fischer[1], Max Schwanitz[1], Reiner Loth[2], Uwe Posner[3], Markus Brand[1],

Friedhelm Schröder[4]

[1] Alfred-Wegener-Institut Helmholtz Centre for Polar- and Marine Research, Centre
   for Scientific Diving at the Biological Station Helgoland, Kurpromenade 211, 27498
   Helgoland.
[2] loth-engineering GmbH, Lochmühle 1, 65527 Niedernhausen, Germany
   [3] 4H- JENA engineering GmbH, Mühlenstr. 126, 07745 Jena, Germany
   [4] Helmholtz-Zentrum Geesthacht, Institut für Material- und Küstenforschung, Max-
   Planck-Straße 1, 21502 Geesthacht

*Correspondence to*: Philipp Fischer (philipp.fischer@awi.de)

**Abstract.** A combined year round assessment of selected oceanographic data and a
macrobiotic community assessment was performed from October 2013 to November
2014 in the littoral zone of the polar fjord systems Kongsfjorden on the west coast of
Svalbard (Norway). A state of the art remote controlled cabled underwater
observatory technology was used for daily vertical profiles of temperature, salinity
and turbidity together with a stereo-optical assessment of the macrobiotic community,
including fish. The results reveal a distinct seasonal cycle in total species abundances
with a significantly higher total abundance and species richness during the polar
winter when no light is available under water compared to the summer months when
24-h light is available. During the winter months, a temporally highly segmented
community was observed with respect to species occurrence with single species
dominating the winter community for restricted times. In contrast, the summer
community showed an overall lower total abundance, as well as a significantly lower
number of species. The study clearly demonstrates the high potential of cable
connected remote controlled digital sampling devices, especially in remote areas, such
as the polar fjord systems, with harsh environmental conditions and limited
accessibility. A smart combination of such new digital "sampling" methods with



classic sampling procedures can provide a possibility to significantly extend the sampling time and frequency especially in remote and difficult to access areas. This can help to provide a sufficient data density and therefore statistical power for a sound scientific analysis without increasing the invasive sampling pressure in ecologically

sensitive environments.

## 1 Introduction

Kongsfjorden (78°55'N, 11°56'E) on the west coast of Spitsbergen (Fig. 1) is described as one of the best studied polar fjord systems in the Arctic (Wiencke, 2004).

The 20–km-long ecosystem opens without a sill in a westerly direction toward the Fram straight (Hop et al., 2002) and is alternatively penetrated by warm saline Atlantic water masses from the West Spitsbergen current, by cold less saline Arctic water from the East Spitsbergen current or a mixture of both (Cottier et al., 2005). This bi-modal hydrographic situation leads to a complex spatio-temporal pattern in

the fjord hydrography with an occasionally more Atlantic and in other instances more Arctic characteristic with respect to the water masses even in the inner fjord system (Svendsen et al., 2002). Due to an increased advection rate of warmer Atlantic water masses in the fjord systems over the last decade (Cottier et al. 2005), the first signs of an overall warming of the fjord system have been observed with an overall decrease

in seasonal ice coverage (Walczowski et al., 2012), significant changes in the phytoplankton community (Hegseth and Tverberg, 2013; Willis et al., 2006), changes in the depth distribution of macroalgae in the shallow waters (Bartsch et al., 2016) and an increase in turbidity due to increased meltwater runoff from the glaciers (Peterson et al., 2002; Bartsch et al., 2016). Although Renaud et al. (2011) and Voronkov et al.

(2013) have recently started to study the food-chain length, trophic levels and the main feeding groups in Kongsfjorden, our knowledge on the temporal and spatial dynamics of the higher trophic levels of the food web is still extremely limited (Stempniewicz et al., 2007). Therefore, important knowledge gaps such as a lack of quantitative data on production, abundance of key prey species, and the role of

advection on the biological communities in the fjord still exist (Hop et al., 2002).

Such knowledge, however, is mandatory for a better understanding of this polar fjord





system and potentially to use it as a model system for future artic change scenarios
under the pressure of global warming. The most comprehensive review thus far on the
occurrence and higher trophic level species in the Kongsfjorden ecosystem has been
performed by Hop et al. (2002) and revealed approximately 34 zooplancton taxa,

between 29 and 396 macrozoobenthos species as well as approximately 30 fish
species in the fjord system in total depending on the type of substratum. Most of these
data have been sampled during intense summer campaigns with ship-supported
sampling methods or by occasional SCUBA diving operations at different sites of the
fjord. Although these datasets are highly valuable, they are mainly restricted to the

polar summer when light is available and sampling can be performed on a regular
basis. A systematic year round assessment of the fjord community, especially of the
shallow water habitats, which are well known as most important as spawning,
hatching and nursery grounds for juvenile specimen (Fischer and Eckmann, 1997b;
1997a; Werner, 1977), is missing.

Thorough assessments especially of higher tropic levels such as fish and
macroinvertebrates are demanding already in northern temperate non-polar waters
because of the required logistics, methods and men-power (Wehkamp and Fischer,
2013c; 2013b; 2013a). In arctic waters with the even harsher conditions with respect
to low winter temperatures, seasonal limited daylight availability and a partly or

complete ice coverage, longer-term and year-round assessments especially in shallow
coastal areas are almost completely lacking. Furthermore, in several hard bottom fjord
system, such as the Kongsfjord system, the shallow-water areas are relatively
inaccessible by trawling with larger vessels due to a complex and highly structured
benthic habitat, with a mixture of rocky bottom and ice-rafted pebbles and stones

(Jorgenson and Gulliksen, 2001). Therefore, most available studies are temporally
restricted to the summer months and the open or deeper water bodies.

In the present study, we present data from a 13-month (October 2013 to November
2014) lasting hydro-biological survey in the sublittoral zone of the Arctic
Kongsfjorden at the southern shoreline close to the research village NyÅlesund at

UMT N 8763194, E 433755 (Fig. 1). With a 2012 installed cabled underwater
observatory (COSYNA@AWIPEV **U**nder**w**ater **O**bservatory - subsequently called
UWO), we continuously recorded the main hydrological parameters temperature,





salinity, pH, Chl-A and turbidity and additionally made a quantitative analysis of the abundance, species occurrence and (for selected species) length-frequency distribution of the fish and macroinvertebrate taxa. For the latter assessment, a stereo-optical macro-biota observatory called "RemOS1" (Remote Optical System) was used

specifically designed for long-term exposure and assessments of fish and macroinvertebrate communities in shallow water areas (Fischer et al., 2007b). Data acquisition was conducted year round, remote controlled with a temporal resolution of 1Hz for the hydrological data and a stereoscopic imaging frequency of 30 min. Parallel to this study, two classic fishing campaign were performed in April 2014 and

September 2014 for 6 weeks each in the same area with standard fyke-nets to provide ground-truth data for the remote-sampled fish data. Because these data were used for several other parallel studies during 2014, they have been published elsewhere (Brand and Fischer, subm.).

The present study aims to demonstrate the high potential of remote controlled sensors to quantitatively assess not only hydrological data such as temperature, current or plankton community with classical CTD probes or VPR recorders but also for the assessment of higher tropic levels such as macro-invertebrates and fish. To the best of our knowledge, there are only a small number of studies and observatories available

world-wide that are trying to assess also higher tropic levels with remote controlled optical systems (Aguzzi et al., 2011; Buckland et al., 2005; Fischer et al., 2007b; Wehkamp and Fischer, 2014) and even fewer in regard to quantitative assessments with respect to specimen's abundances and species specific length-frequency analysis in an area. Because these technologies will certainly develop and improve over the

next years, this study also discusses certain specific requirements and challenges for such systems, especially for shallow water artic areas.

## 2 Materials and Methods

The UWO was built up in 2102 in the framework of COSYNA (Coastal Observing

Systems of the Northern and Arctic Seas). The system comprises a land based FerryBox system equipped with various hydrographic sensors (Table 1) receiving water from a remote controlled underwater pump station at 11 m water depth. Additionally, a cable connected (fibre-optic and 240V power) underwater node (Fig. 2) was installed close to the pump station in 11 m water depth providing power (48V)



and network (TCP/IP 100Mbit) connection for additional *in situ* sensors. To install or exchange sensor equippmemt at the node system by divers, the node is equipped with four underwater matable power/ethernet connectors and two additional underwater matable power/rs232 connectors.

For the experiment described in this study, the node system was equipped with an upward looking ADCP positioned in 13-15 m water depth (depending on the tide cycle), a SBE38 temperature sensor positioned in 11-13 m water depth (depending on the tide cycle) and a vertical profiling sensor carrier. The profiling sensor carrier was fully remote controlled via the Internet and was operated year round from October

2013 to November 2014 from Germany. It was equipped with a CTD for the assessment of the main hydrographical parameters and the stereo-optical camera system RemOS1 (Fischer et al., 2007b; Wehkamp and Fischer, 2014) for macrobiota assessments. Using the stereo-optic sensor, we assessed the macrobiota, jellyfish and fish community along the vertical depth profile from 11m water depth to the surface

wit the sensors looking from a distance of about 2.5m towards a steep wall that reached from 11 m of water depth to 3 m below the mean sea level (Fig. 2). The upper part of the wall was dominated by brown algae of the type of Alaria esculenta, the lower part by *Saccarina latissima* and the two red algae species *Phycodris rubens* and *Ptilota gunneri*. Using the vertical profiling unit, we conducted a one-year continuous

stereo-optical survey of the fish and the macrozoobenthos community in five depth strata (11-9 m, 9–7 m, 7-5 m, 5-3 m and 3 m-surface). The stereo-optical system and the CTD probe were remotely positioned every day between 11:00 and 13:00 hours in one of the five depth layers with the exact depth being calculated as distance from the bottom. This means that the effective water depth changed with the tide cycle for

max. 1.5 m, but the system itself had a fixed position above ground (1 m distance from the bottom for the depth stratum 11-9 m, 3 m distance for the depth stratum 9 -7 m, 5 m distance for the depth stratum 5 – 7 m, 7 m distance for the depth stratum 3 – 5 m and 9 m distance for the depth stratum 3 – 0 m). The daily target depths were selected randomly for each week such that all of the depth strata were sampled once

per week for 24 h. Missing depth, e.g., because of system or connection problems to the underwater observatory were repeated on the weekend. The system was positioned for 24 h at the selected depth stratum and made stereoscopic images every 30 min. Parallel, all other *in situ* and ferrybox sensors recorded with a frequency of 1 Hz. The image pairs and all the hydrographic data were transferred automatically via Internet



to Germany for further daily processing. All hydrographic data were automatically quality-controlled by automated procedures, flagged as good, probably good and bad and stored at a central data server in Germany, Geesthacht under an open access policy at http://codm.hzg.de/codm/. For our study, only the data with the quality flag

probably good and good were used. Based on these data, we analysed the temporal succession of the shallow water fish, jellyfish and macrozoobenthos community in this kelp dominated shallow water arctic habitat in Kongsfjorden. Organisms on the stereoscopic images were analysed in a two-step procedure following the routines described in Wehkamp and Fischer (2014). The 48 stereoscopic image pairs of each

day were first scanned manually for the presence of organisms. This scanning was performed with an image analysis software that presented the left image of the stereoscopic pair for at least 5 seconds on a 21" high resolution computer screen. Only two persons did this basic analysis step over the entire year and thoroughly counterchecked their object findings. During this first step, all the specimen found on

an image were counted and pre-classified in the categories fish, jellyfish, apendicularia, pelagic crustacean, benthic crustacean, pteropods and chaetognats. Organisms that could not be classified in one of these categories were classified as "others". The analyser (person who did the analysis) had the possibility to increase or decrease the image brightness or to enhance the contrast by a single mouse click

quickly. The possibility for such a rapid pre-processing of the 48 stereoscopic image pairs revealed to be most important because 48 image pairs were produced every day year round. This rapid assessment procedure allowed a first analysis of all the images per day within approximately 15 minutes, so that a quasi-online overview over the actual situation under water in the target area and on the functioning of the monitoring

system was achieved within 24 hours. With this procedure, problems of the system itself or with the data transfer could be detected fast and could be addressed and solved. With this daily rapid assessment routine, we could achieve an acceptable level of operational stability of the systems with less than 15 unplanned offline days over the entire sampling period of 13 months. Unplanned offline days occurred mainly due

to failures in the land based power support system. During such phases, the underwater part of the system was shut down to avoid hardware damage due to spontaneous and possibly critical voltage fluctuations.

In a second image analysis step, all the images where organisms were detected were rectified, which means that the geometry of the images was corrected to eliminate



image distortions due to the lens of the camera. This correction was performed with the modified MATLAB routine "stereo_gui" (Wehkamp and Fischer, 2014). After this step, all the objects that were detected in the first image analysis step were measure (standard length in fish, carapax length in macrocrustacea and max.

dimension in all other organisms) and identified as precise as possible, i.e., to species level in most fish species except for the two cod species *Boreogadus saida* and *Gadus morhua*, which were not distinguished properly on the images. Furthermore, amphipoda or apendicularia were only identified to the class level.

Because we had a clearly restricted water volume that was assessed by the camera

system (volume between the camera and the vertical wall) we calculated the "catch per unit effort" of the system by summarizing all the individuals found on the images per 24 h and depth stratum. These CPUE × 24 h$^{-1}$ data were used as basis for all further calculation. We did not recalculate these data on a defined water volume (which is possible) to avoid confounding calculations between benthic organisms

living on the two-dimensional bottom or the surface of the algae and planktonic organisms living in the three-dimensional water column.

Length-frequency measurements on the 3D-image pairs were performed pooled for each month for the cod species (mainly *Gadus morhua*), the common sea spiders (*Hyas araneus*), the two main jellyfish species (*Beroe sp.* and *Aglantha digitale*), the

apendicularia and the pteropods (*Clione limacina*). For these species, all the organisms were measured except for the month when more than 200 specimens occurred within one month. In this case, only 200 specimens were measured by randomly selecting over the day of the month.

**3 Results**

**3.1 Habitat description**

The Kongsfjorden shallow water ecosystem is characterized by large kelp beds of different species of macroalgae (Bartsch et al., 2016) between 0 and approximately 12-15 m water depth. The site where the observatory has been set up is, therefore, characteristic for the fjord habitat and provides a highly diverse habitat

with a steep wall completely covered with large macroalgae followed by a sandy to muddy slope that begins at approximately 11 m water depth at the base station





of the observatory. The five depth layers covered by the stereo-optical camera system covered the typical vertical gradient of a littoral habitat with a surface near pelagic habitat (depth range 0-2 water depth (Fig. 3a), a typical lito-pelagic habitat close to the upper edge of the drop off (2-4 m water depth (Fig. 3b), the upper

drop off edge between 4 and 6 m water depth) with dense horizontal and vertical macrophyte coverage (Fig. 3c), the vertical wall of the drop off with overhanging structures and grotto like crevices (water depth 6-8 m, Fig. 3d) and, finally, the lower edge of the drop off where the wall goes over in the typical benthic habitat with a gentle slope formed by sand and mud in a deep of around 11 meters,

decreasing further towards North to the center of the fjord (Fig. 3e).

The observatory technology allowed for daily vertical CTD profile every noon at approximately 12:00 hour with a sampling frequency of 1Hz at a constant profiling speed of 1.5 m per minute from approximately 10 m water depth (depending on the tide) to 1 m below the surface. The Ferrybox unity additionally provided

complementary hydrographic data from a fixed water depth of 11 m. Fig. 4 shows the compiled data for water temperature (°C), salinity (PSU) and turbidity (FTU) from October 2013 to November 2014. The data reveal a distinct seasonal cycle in the water temperature with the lowest values of approximately -1.0°C in the winter months from October to April and the highest temperatures up to approximately 8°C

during the summer months, May to September. Most interestingly, however, are the distinct short term changes in water temperatures even within the individual seasons. These changes spanned ranges of up to 4°C within shortest time periods of few days both in the summer an in the winter. While the average water temperature, for example, during mid of December to the end of January was between -0.5 to +0.5°C,

the water temperatures then suddenly increased within few days up to 3°C and stayed at this comparatively high level until end of March, when it dropped again to approximately 0.5°C again. In May, the temperatures increased again and reached the highest values of up to 7.7°C in the surface layers, which indicates a distinct stratification during this time. In July to September, this stratification dissolved, and

the water temperatures were almost equally distributed over the water column. Similar temporal patterns were observed also in salinity (Fig. 4), which indicates that the overall patterns in the water temperature in the shallow littoral zone of the fjord system were also significantly determined by a fast (within days) exchange of water



masses that brought either colder and lower saline arctic water or warmer higher
saline water masses even to the shallow fjord areas.

Fig. 4 shows the seasonal patterns in turbidity over the water columns. The data
indicate that the overall turbidity significantly increased during the seasonal cycle

with higher values from July to September and low values during the rest of the year.
However, Fig. 4 also shows a longer lasting local and distinct increase in turbidity
close to the bottom in May and June. These high turbidity values during this time are
confirmed by both systems, the vertical profiling *in situ* probe as well as the Ferrybox
unit.

**3.2 Species community**

Figure 5 (upper panel) shows the sum of individuals counted on the images per week
for the month October 2013 to November 2014. The average values and standard
deviations per month were calculated based on four or five weekly CPUE values
depending on how many weeks a month had. The analysis revealed a distinct seasonal

cycle with high specimen abundances during the winter months from December to
April, lowest values from May to July and a second smaller peak in August and
September. Fig. 5 (lower panel) shows the same monthly abundance values but
separated by groups of organisms. Ten different groups of organisms were identified
over the year, namely, apendicularia, benthic crustacea, birds, chaetognaths, fish,

jellyfish, molluscs, pelagic crustaceans, polychaets and pteropods. From these groups,
six occurred in higher abundances, at least, during a certain phase of the year (benthic
crustacean, fish, jellyfish, apendicularia, chaetognaths and pteropods).

During the winter – spring peak, benthic crustaceans had the highest share of the total
species abundances followed by jellyfish, pteropods and fish (Fig. 5, lower panel). In

contrast, the summer – autumn peak was almost completely formed by apendicularia
and a smaller share of fish.

When analysing the winter-spring phase (December – March) and the summer-
autumn phase (August – October) separately and in detail, a strong spatial separation
of the winter-spring and summer-autumn community emerged with respect to the

position in the water column (Fig. 6). While the overall share of the winter-spring
community was benthic or benthic- associated except for the jellyfish; this benthic-





associated community was almost completely missing in the summer and autumn, except for a small share of fish.

Except for apendicularia, all of the other highly abundant species were identified to the species level if possible. Fig. 7 shows the species composition of benthic

crustaceans (upper panel), fish (middle panel) and jellyfish (lower panel). The analysis revealed that approximately 90% of the benthic crustaceans identified over the year were made up by a single species, the great spider crab *Hyas araneus* (L.). Besides, also hermit crabs (Paguridae) were found occasionally as well as benthic living decapod crustaceans, which most probably belonged to the mysid species *Mysis*

*oculat* (approximately 10% share). *Hyas araneus*, however, clearly dominated the benthic decapod community especially in the winter month of February when a mass invasion of this species was observed in the area.

A similar uniform pattern was observed in fish (Fig. 7 – middle panel). 81% of the fish on the images were classified as cod of either one of the two species: *Gadus*

*morhua* (L.) (50%) or *Bodeogadus saida (L.)* (31%). The differentiation of this two species, however, has to be perceived critically because it was based on coloration, which is especially problematic in young specimens. For all the subsequent analysis, we pooled these two fish species and summarized them under "Gadidae".

The most diverse groups over the year were the jellyfish (Fig. 7 – lower panel). A

total of nine different species plus one class "unidentified" were found. Integrated over the year, the most dominant jellyfish species (57%) belonged to the group *Beroe* sp. followed by *Aglantha digitale* (8 %) and *Pleurobrachia pileus* (5 %). All the other identified species (*Physonectidae* sp., *Mnemiopsis leidyi*, *Mertensia ovum*, *Euplocamis dunlapa, Cyanea* sp., *Bolinopsis iunfundibulum* and *Aglantha digitale*)

occurred in abundances with a total share of < 1%. Unfortunately, 37% of the jellyfish could not be clearly identified to the species level and, therefore, had to be left unidentified. These species did most certainly not belong to the above mentioned identified species, which indicate that the jellyfish diversity in this area is even higher. For the dominant species of the six major biota groups (benthic crustacean, fish,

jellyfish, apendicularia, chaetognaths and pteropods), the body sizes were measured for up to 200 randomly selected specimens per month (if available). In benthic crustaceans, the carapax length from the tip of the rostrum to the end of the telson (in a normal body position) was measured; for fish, the standard length; for jellyfish, the





largest body dimension (either longitudinal or transversal), and for chaetognaths and pteropods, the longitudinal body axes was measured. The system allowed for an accuracy in length measurements of approximately 3% (Wehkamp and Fischer, 2014). Fig. 8 to 10 show the size-frequency distributions of the six measured groups

per month over the seasonal cycle from October 2013 to November 2014. As most abundant species during the winter months, November to March, *Hyas araneus* showed an average carapax length between 50 and 100 mm (Fig. 8 – upper panel) with no temporal trend over the months. However, in November and December 2013, also larger animals with a carapax length up to 180 mm appeared in the area, which

disappeared during the spring and re-appeared again one year later in November 2014.

In contrast, in the pooled species group "Gadidae", a clear increase in the average length over the months was observed (Fig. 8 – lower panel). Starting in November 2013, the YOY cohort appeared in the area with an average standard length between

70 and 100 mm. This 2013 cohort stayed in the area until March 2014 when they reached an average length between 100 and 125 mm. After this time, no more cod was observed in the area over the spring and summer until then next YOY cohort appeared for a short time in higher abundances in August 2014 with an average standard length between 40 and 70 mm (mean ± sd = 65 ±16 mm). After this time, no

more YOY cod could be observed in the shallow area. Instead, larger cod of up to 300 mm were observed sporadically in the shallow waters (Fig. 8 – lower panel, September – October 2014).

All of the other species that occurred in higher abundances in the shallow areas around NyAlesund belonged to the pelagic community. In jellyfish, the ctenophore

*Beroe sp*. made up a major share of the planktonic community and appeared with higher abundances in the winter months, November to April, but with only few specimen during the summer months. For *Beroe sp.,* no temporal size distribution pattern was observed over the months (Fig. 9 – upper panel). The highest abundances were observed in February with an average size in longitudinal direction of 45 mm

spanning from 10 mm to 75 mm with average values of 32 ±8 mm (mean ± sd). Jellyfish occurred with the highest abundances in the shallow-most water layer between 0 m and 2 m and in only lower abundance in the water columns between 2 m





and 8 m. In the deepest water layer close to the bottom, the abundances of *Beroe sp.* were the significantly lowest ober the entire water column (LRχ² = 105, df = 3, p < 0.001).

Another temporally dominant but more agile species compared to the jellyfish were

the chaetognaths. This group occurred with the highest abundances also during winter months (Fig. 9 – lower panel) and were also completely missing during the polar summer. Compared to the jellyfish, however, which were almost equally distributed over the water column except for the deepest stratum, Chaetognath occurred highly stratified in the water columns with the highest abundances in the 2-4 m depth layer;

no specimen was found in the surface layer shallow than 2 m, and significantly lower abundances were also found in the deeper water layers (LRχ² = 490, df = 3, p < 0.001). With lengths between 20 and 50 mm (mean ± sd = 32 ±8 mm), chaetognaths formed a major part of the pelagic winter community in the shallow areas. A detailed image based on species identification as well as on the size distribution of the

observed chaetognaths suggest that the majority of the observed specimen belong to the species *Parasagitta elegans* (Verrill 1873).

Temporally, almost synchronized with the chaetognaths, also pteropods (Fig. 10 – upper panel) occurred in the water column and were observed in higher abundances until April. On the images, only *Clione limacina* was observed with body sizes from

10 to 40 mm and a mean size of 23.1 ±5.5 mm (mean ± sd). Similar to the above described chaetognaths and jellyfish, also *Clione limacine* occurred highly stratified in the water column with a peak abundance in the 2 – 4 m depth layer and significantly lower abundances both, in the surface layer and in deeper water strata (LRχ² = 143, df = 4, p < 0.001).

The only species that reached higher abundances not in winter but during the summer months were the apendicularia (Fig. 10 – lower panel). Especially during the month August to October a mass invasion of apendicularia in the upper water columns was observed. As for the other pelagic species, those higher abundances were mainly observed in the 2 to 4 m water layer while no apendicularia were observed in the

uppermost layer close to the surface and significantly lower abundances were observed below 4 m water depth (LRχ² = 1039, df = 3, p < 0.001).



## 4 Discussion

Shallow water areas are well-known as important habitats for shallow water
communities (Reyjol et al., 2005). Due to the often higher structural complexity of
shallow coastal waters compared to the deeper parts of the ocean, coastal habitats are
often observed as important spawning areas and nursery grounds that form the
biological backbone of a diverse and stable benthic and fish community in the
associated marine habitats. For the same reason, however, studying higher tropic biota
in coastal environments is challenging in regard to a detailed assessment of their
temporal and spatial dynamics especially of mobile communities. The high structural
complexity, especially of shallow water hard bottom or reef habitats, often prevents
classical ship-supported and space-integrative sampling methods such as trawling or
box coring (Brickhill et al., 2005; Fischer et al., 2007a; Wilding et al., 2007).
Assessments in these structurally complex environments often require small-scaled
and highly specialized "sampling" methodologies often based on optical mapping or
imaging technologies operated by divers or ROV's, depending on the water depth.
Brickhill et al. (2005), Fischer et al. (2007b) and Wehkamp and Fischer (2014)
discussed the potential of such techniques specifically for the assessment of fish-
habitat relationships in temperate and boreal habitats such as the Southern North Sea.
They concluded that the in these waters, the comparatively restricted transparency of
the water, the lower water temperatures and the harsher weather conditions often lead
to only short operation times that result in low numbers of freeze-frame sub-samples
taken in most studies, preventing a thorough analysis of the species-habitat
relationships due to an in sufficiently fine scale sampling frequency. These limiting
factors, especially of diver-operated *in situ* video technologies, often lead to
extremely high variability in organism counts per frame with too many zero counts,
especially when the target organisms are mobile. This leads to a dramatic loss of
statistical power in the subsequent data analysis (Brickhill et al., 2005).

These limitations are even more distinct in polar areas where the diver-supported
access to the ecosystem is both temporally restricted and extremely expensive.
Sampling structurally complex coastal habitats in polar areas is often only possible
during a restricted period of time in the polar summer when light is available and the





temperatures allow for *in situ* methods. Therefore, our knowledge on polar shallow
water ecosystems and especially their role as nursery and juvenile habitat is extremely
restricted. Most of the recent studies (e.g., Hop *et al.* 2002; Svendsen *et al.* 2002; Hop
*et al*. 2006) in our addressed study area have been conducted during summer, when

the ford system is accessible by research vessels. Although the summer productive
period is of great importance for Arctic ecosystems, several crucial processes (e.g.,
reproduction) take place during other seasons and especially during the polar winter.
During these times, however, almost no information is available in most arctic fjord
systems (Kwasniewski, 2003). Understanding polar ecosystems in the context of

global warming and expected or already observed ecosystem changes (Bartsch et al.,
2016) is, however, crucial for thoroughly understanding the ecosystem behaviour in
polar areas.

In this study, we did not provide results from experimental work in Kongsfjord based
on discrete studies with a clear short term ecological hypothesis. In contrast, we

provide data from a one year long quantitative assessment of hydrographic parameters
together with quantitative data on the macrobiota community assessed by a remote
controlled cable-connected underwater observatory installed in a typical shallow
water habitat in the Kongsfjord. Using a remote controlled vertical profiling system,
we were able to continuously assess temperature, salinity, turbidity and other

hydrographic parameters together with the shallow water macrobiotic community
over the entire water column from the benthic over the epi-benthic to the pelagic
realm in a high temporal resolution. To our knowledge, this is the first dataset from
Kongsfjord that reveals such a year round assessment of the shallow water
macrobiotic community together with the quantitative data of the water temperature,

salinity and turbidity and, therefore, allows a deeper insight in the coupling of the
seasonal dynamics of the biology and the hydrography compared to pure summer
studies. The data revealed a distinct winter community in the fjords shallow water
ecosystem, which by far exceeds the summer community in both, abundance and
species diversity. Although we have not yet calculated biomass per m$^3$ for the

assessed species, our data clearly showed that the species abundance and species
richness is highest during the polar winter that begins in December when no more
light is available under water. During this time, except for the apendicularia, most
species including fish (mainly gadids of the species *Gadus morhua* and Boerogadus



saida), jellyfish (mainly *Beroe sp.*), Chaetognath (*Parasagitta elegans*), pteropods (*Clione limacina*) and smaller benthic and epi-benthic crustacean (most possibly *Mysis oculata*, C. Buchholz pers. comm.) invade the shallow water zone and build up highest abundances. During this study, an overall peak abundance was observed in

February when the common sea spider *Hyas araneus* clearly dominated the community in numbers and biomass for a short time. Only one month later in March, however, *Hyas araneus* almost completely disappeared when fish, jellyfish and pteropods formed the predominant community with respect to the overall abundances. The "winter" community persisted until April and then almost vanished. The time of

the winter community "disappearance" highly corresponds with the increasing availability of light under water. Although sun light is available at NyAlesund again already during mid of March ([https://www.yr.no/place/Norway/Svalbard/Ny-%C3%85lesund_observation_site/](https://www.yr.no/place/Norway/Svalbard/Ny-%C3%85lesund_observation_site/)), the inclination angle of the light is still low until April so that only a small fraction of the sunlight is penetrating the water columns

(personal observation). However, to really correlate the presence of the "winter community" with the availability of light underwater, discrete measurements of the light intensity and light quality are necessary in the different depth strata to reveal if light is an ultimate factor for the temporal occurrence of the fjords shallow water winter community or only a proxy associated with other environmental factory. Our

data suggest that especially water temperature may also have a significant influence on the spatio-temporal occurrence of the winter community. Our daily sampled temperature profiles clearly showed that water temperature in the shallow water areas of Kongsfjord can change within short times, even in winter, between < 0°C and up to 4°C. In particular, the peak abundance in the common seas spider *Hyas araneus*

corresponds with the time of higher water temperature during February, and the collapse of the spider abundance occurred when the water temperatures decreased from 4°C to only approximately 2°C again. Similar temporal pattern could be observed also in the overall species abundance in April when a short cold phase in the water temperature occurred. However, this seemingly corresponding changes in the

biotic community and the changes in the abiotic environments also may be purely by chance and we do not know yet if there are functional relationships between these observations. The permanent installation of the cabled underwater observatory at NyAlesund allows to formulate and test such a hypothesis of a persisting shallow water "winter community" in the fjord system as well as the hypothesized controlling



or at least affecting abiotic factors.

Our data additionally reveal another distinct community during the summer months when the temperatures increased up to 8°C in the fjord. Then, appendicularia occurred in higher abundances for a restricted time, i.e., from August to October, in the shallow
water with a peak in abundances in September. In contrast to the winter community, which was mainly benthic or at least benthos-associated, this summer community was almost completely dominated by a single appendicularia species, most certainly belonging to the genus *Oikopleura sp.* (Dahms et al., 2015).
Besides appendiculria, also juvenile cod fish were found in September in the deeper
littoral water layers closely associated to benthic habitats. The detailed length frequency analysis of this cohort revealed that these fish were the YOY offspring of the same year (YOY cohort 2014) with an average standard length of $65 \pm 16$ mm. The data also revealed that these fish seem to stay in the littoral zone (even though the overall abundances strongly decreased over winter) and continuously grow and reach
an average standard length of 100 to 125 mm in February – March at age-class 1 when they seem to quantitatively leave the shallow water habitats. This outcome indicates a complex migration pattern of YOY cod in this area with a short winter phase in the littoral zone of the fjord system of Spitzbergen and a later migration towards deeper or offshore habitats as adults. Such temporally restricted shallow
water phases have been observed already for several other cod species, especially during their juvenile phase (Pihl, 1982) and assumed a juvenile behaviour to prevent predation by older conspecifics in the deeper adult habitats (Ruiz et al., 1993) as well as an improvement in foraging efficiency of the juveniles during their non-piscivore microzoobenthic benthic feeding phase (Pihl, 1982).

In contrast to the clearly visible seasonal growth pattern in the cod species, no distinct growth could not be observed in none of the other species even in the highly abundant common sea spider, which showed a persisting size range between approximately 50 and 80 mm during the entire winter month except for the month of November in both years when larger animals between 120 and 180 m were observed in the area, even
though in much lower abundances.

As clearly stated before, this study does not provide a singular hypothesis driven question; instead, it focuses on a basic assessment of the temporal (and with respect to



the water column also spatial) pattern in the macrobiota community distribution and possible hydrographic factors that influence the shallow water biota. The results of this study are by far incomplete and only represent a one-year study at a specific site in the Kongsfjord ecosystem, which may or may not be representative of the shallow

water community of this area. However, the study presents a continuous year round data set in a temporal resolution of one week, which is, to our knowledge, not available in any other fjord system and especially not in the arctic environment where winter data are missing at almost every level. However, even though the data provide a unique year round insight in a polar shallow water fjord community, we can assume

that the technology used here has a certain bias with respect to species selectivity. Therefore, these data have to be taken with care. For instance, comparing our stereo-optically assessed fish data with data from classical sampling devices in Kongsfjord (Brand and Fischer, subm.; Hop et al., 2002; Renaud et al., 2011) or even with sporadic divers observations (Brand and Fischer, subm.; Hop et al., 2002), it becomes

clear that also our optical sensors are species selective. (Brand and Fischer, subm.), for example reported for the summer month a distinct occurrence of the benthic sculpin Myoxocephalus scorpius, a typical temperate and highly camouflaged benthic fish species in fyke-net catches. Although we detected *Myoxocephalus scorpius* during summer also on the stereoscopic images, the overall abundance remained quite

low. Unfortunately, the fyke net catches of Brand and Fischer (subm.), as most other available marine studies of the fjord, are only available for the polar summer months when our stereo-optical data revealed the lowest overall biota abundance at all. However, taking into account that fyke-nets are highly time integrative and catch fish only directly at the bottom, the fyke-net and optical data may be rather

complementary than contradictory. In the study of Brand and Fischer (subm.), fyke nets with a mesh size of 12 mm and a steering net of 18 mm were used. This type of net gear is highly selective for strictly benthic fish species with a high potential of entanglement, such as sculpins. In contrast, a stereo-optical method is most probably less selective for benthic highly camouflaged fish species and may significantly

underestimate those. Instead, our overall image assessment procedure was thoroughly performed by two different persons and showed similar results with respect to the quantitative detection of even small benthic mysids. Therefore, we assume that we would have detected also sculpins if available in higher abundances and thus conclude that the quantitative relation of the average abundance between the major fish species



found on the images might be more precise as found in the fyke net catches. This outcome seems to be supported also by the available diver observations in that area at least during summer. Hop et al. (2002) and Renaud et al. (2011) both reported the cod species *Gadus morhua* as one of the most abundant species in the area which would

be in accordance with our findings. Nevertheless, the comparison of these two methods show that there is a large uncertainty with respect to the methodological approach that should be used in future studies. Furthermore, our *in situ* optical methods allow for a low-invasive abundance estimate, for a precise length-frequency analysis of the mapped fish and also for a continuous year round assessment of the

community. However, it does not allow for further investigations such as stomach content analysis and precise aging based on scale or otolith analysis. If we manage to combine such continuous hydrographic and community observations using cable-connected observatories with classical ground trouthing fishing or sampling methods, we may reduce our scientific fishing effort to a limited number of specimen, which

are needed for specific detailed analysis such as stomach content and otolith-based aging and obtain the required more invasive stock abundance and growth data via non-invasive optical methods. These approaches may finally enable the reduction of our fishing effort without losing the required data density and therefore contribute to the increasing scientific demand of a resource conservative science also in fish and

community ecology, especially in ecological sensitive areas, such as the polar fjords or marine protected areas.

**Competing interests**

The authors declare that they have no conflict of interest.

**Acknowledgments**

We express our strong thanks to the AWIPEV staff, i.e. Rene Buergi and Verena Mohaupt who made the continuous opperartion of the underwater observatory in this remote side possible. We furthermore want to thank the numerous divers from the

AWI dive group who did a great work during our maintenance missions as well as María Algueró Muñiz and Cornelia Bucholz for species identification of the jellyfish and the mysids. Special thanks also go to Christian Wiencke who stongly supported



the idea of a cabled underwater observatory at AWIPEV in the initialisation phase.
This work has been supported through the Coastal Observing System for Northern
and Arctic Seas (COSYNA).



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



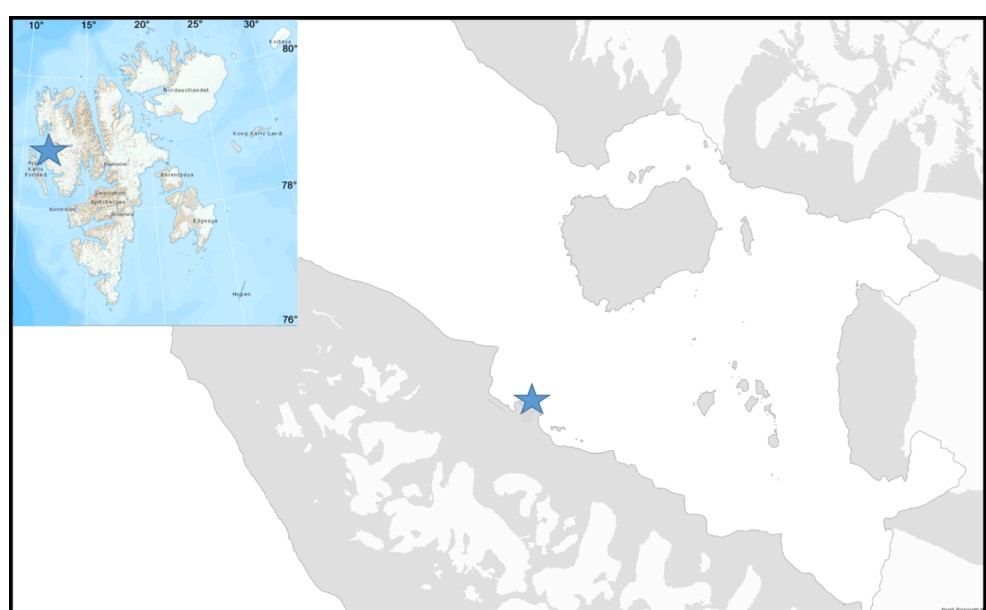

Figure 1: Spitzbergen with Kongsfjorden (★) in small inlay panel upper left corner) and the location of NyÅlesund in Kongsfjorden (★). Source: Norwegian Polar Institute (2014). Kartdata Svalbard 1:1 000 000 (S1000 Kartdata). Norwegian Polar Institute https://data.npolar.no/dataset/63730e2e-b7a6-4d14-b341-c661ccdc5254.



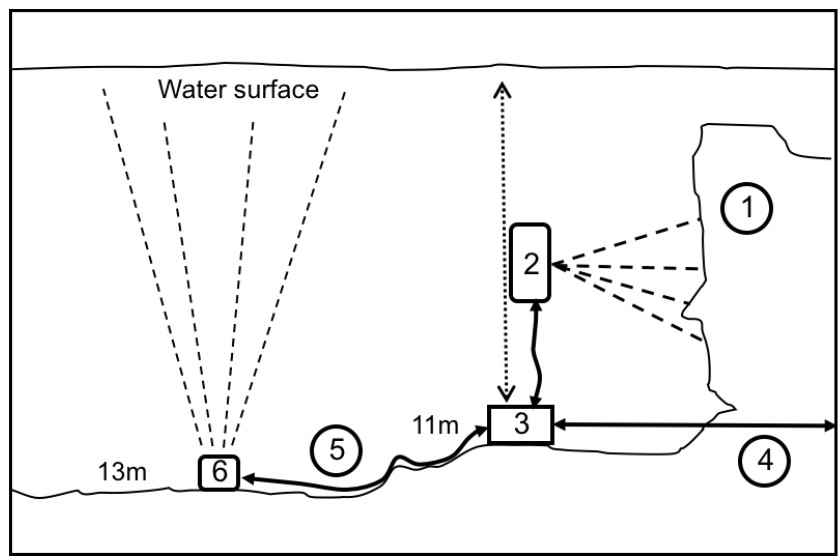

Figure 2: Sketch of the underwater installations with the underwater base station and the vertical profiling unit off NyÅlesund. Numbers refer to num-bers in in sketch: (1) Steep wall (drop-off) with vertical zonated macro-phyte coverage; (2) Vertical profiling sensor carrier with CTD and stereo-optical imaging device (RemOs1) looking towards the wall; (3) Underwater node with wet-matable plugs; (4) Combined power/fibreoptic cable to land; (5) Combined power/rs232 cable from node to ADCP; (6) ADCP. For details on the single components, see text.



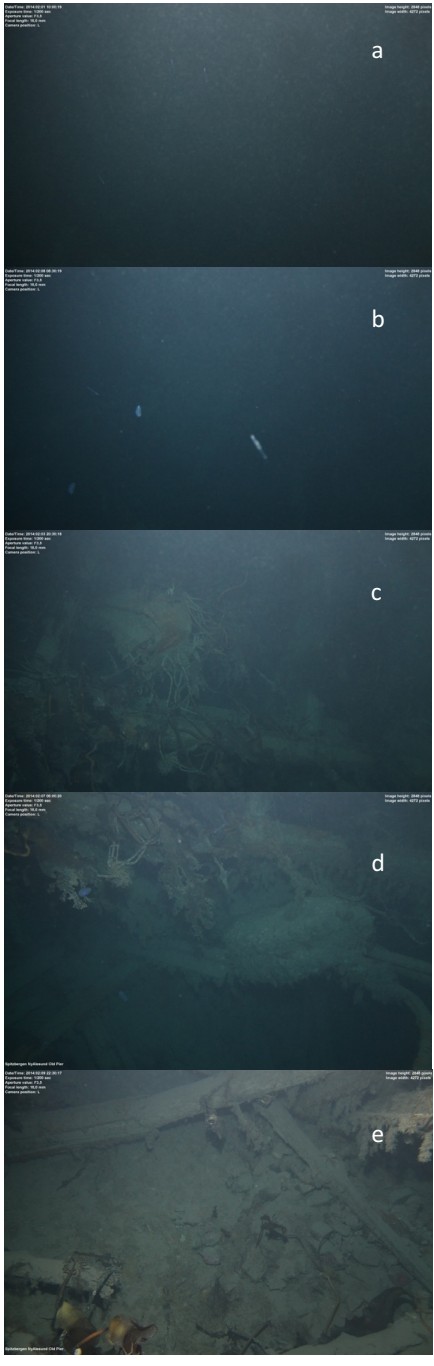

**Figure 3 a,b,c,d,e: View of the stereo-optical system RemOs1 in the five differtent depth strata.. Panel a: depth stratum 0-2 m, panel b: depth stratum 2-4 m, panel c: depth stratum 4-6 m, panel d: depth stratum 6-8 m, panel e: 8-11m.**



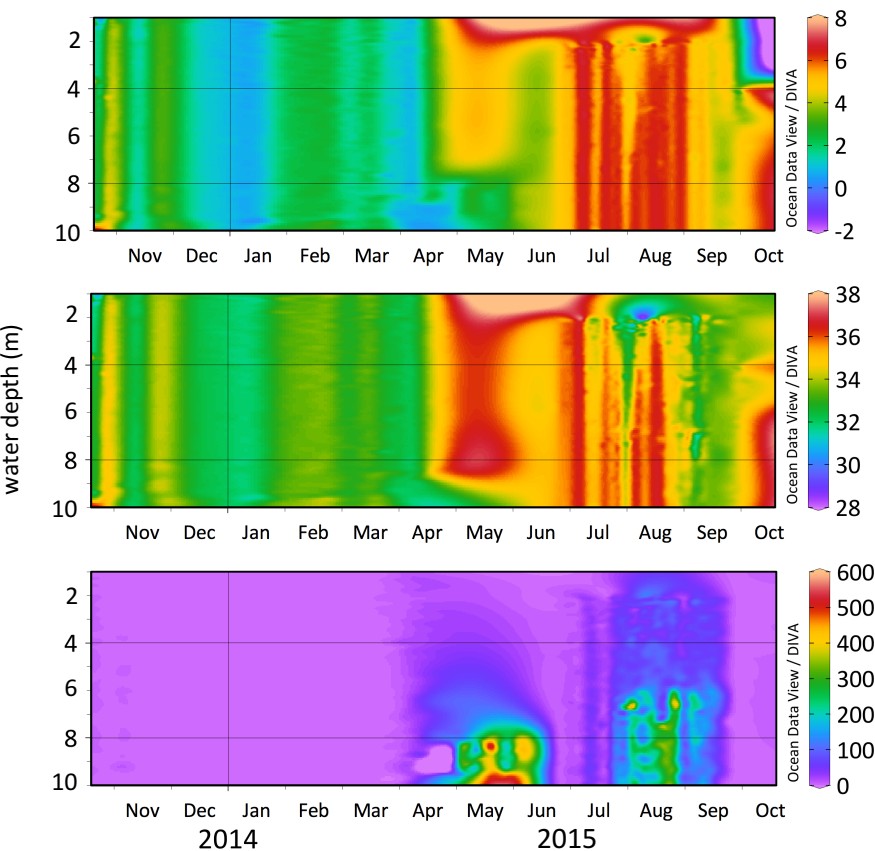

**Figure 4: Temporal - spatial pattern in water temperature (°C – upper panel), salinity (PSU - central panel) and turbidity (FTU – lower panel) from October 2013 to October 2014 for the depth range 1 to 11 m based on daily vertical CTD profiles from 10 to 1 m and the FerryBox data from 11m (fixed inlet).**





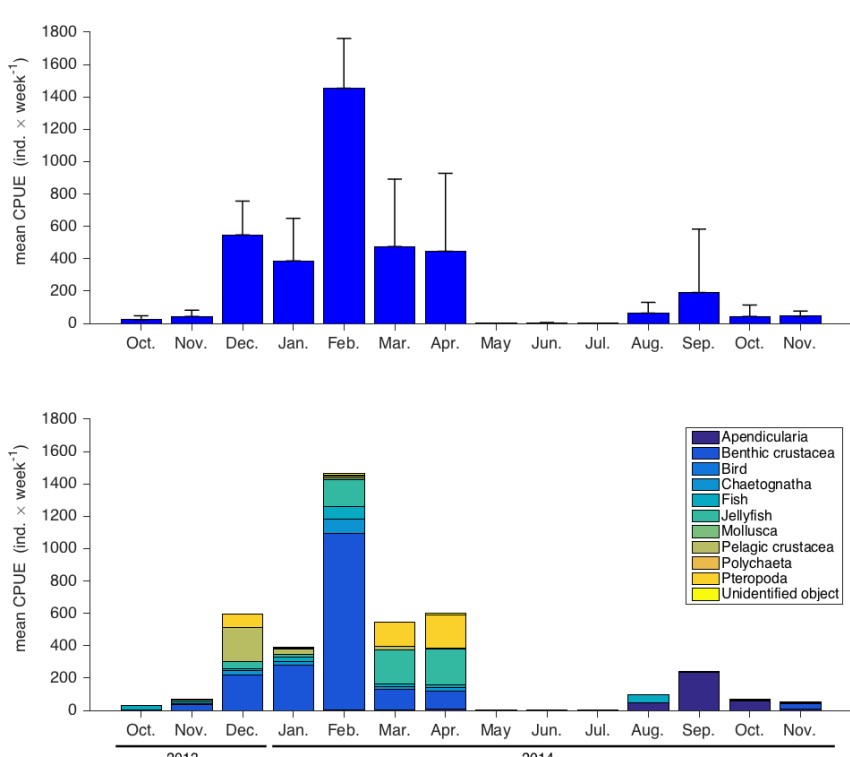

**Figure 5: Seasonal cycle in total species abundance (upper panel) and species composition (lower panel) pooled per month of the year. For detail with respect to „Catch per unit effort" see text.**



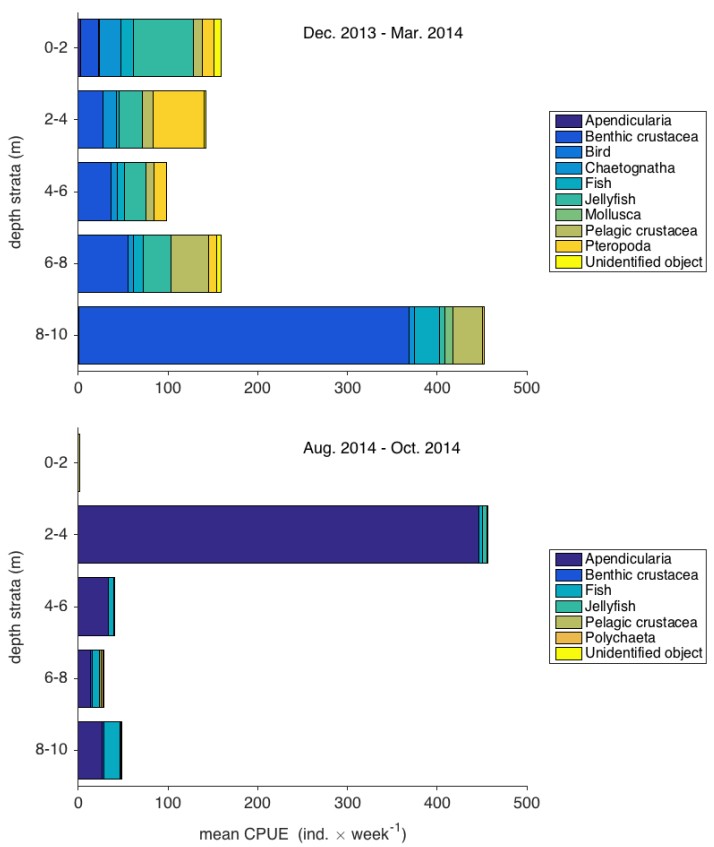

**Figure 6: Vertical distribution of the different species groups over the water columns. For details with respect to „Catch per unit effort" see text.**



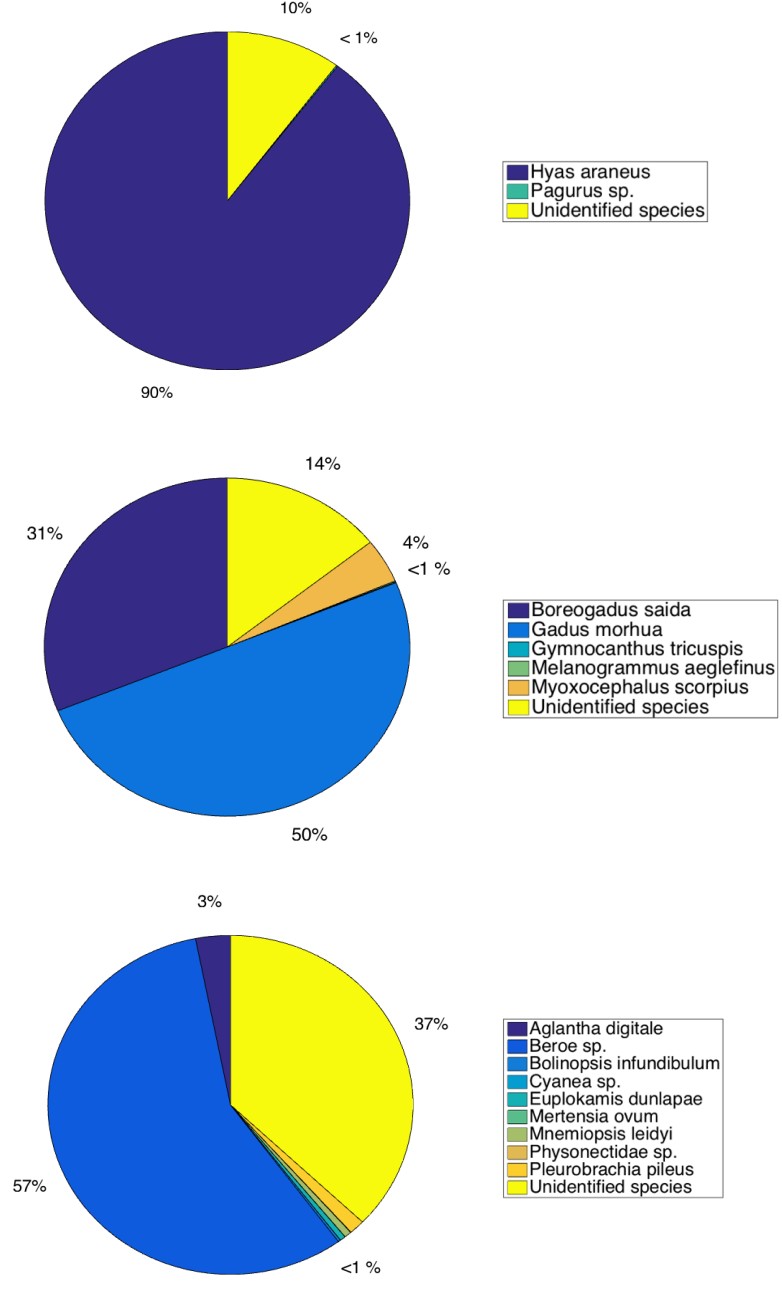

**Figure 7: Percent distribution of the different species within the
different biota groups. For details see text.**



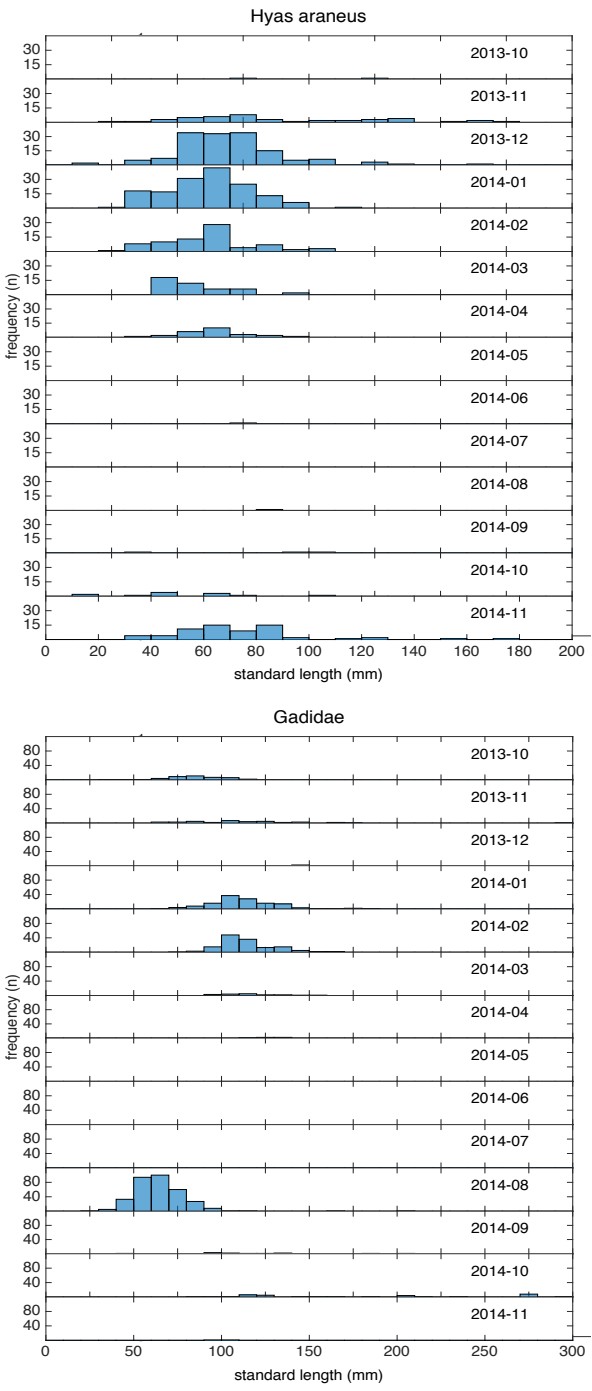

**Figure 8: Length frequency distributions of selected species resp. species groups (see panels) over the seasona cycle.**

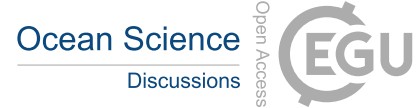

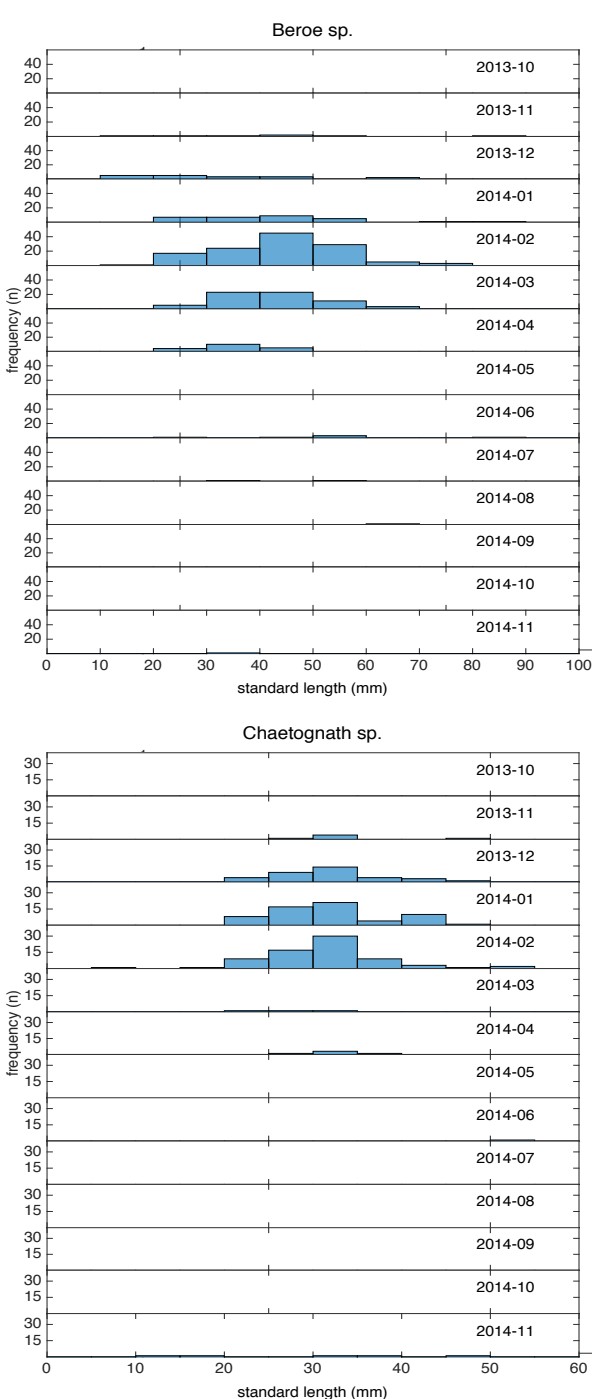

**Figure 9: Length frequency distributions of selected species resp. groups (see panels) over the seasona cycle.**





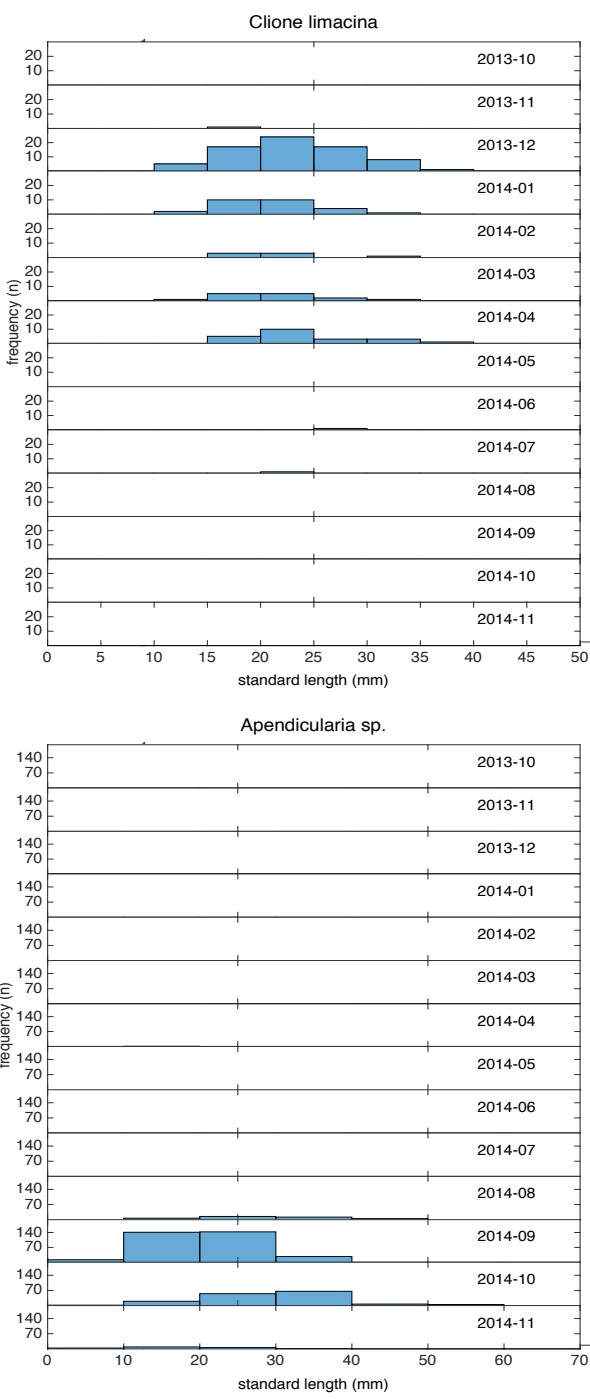

**Figure 10: Length frequency distributions of selected species (see panels) over the seasona cycle.**



**Table 1: Sensors attached to the COSYNA@AWIPEV UWO at UMT N 8763194, E 433755. The Ferrybox has its water inlet at a fixed depth of 11m below mean sea level (http://vannstand.no/index.php/nb/english-section/sea-level-data). The RemOs1 system is profiling from 11m water depth to the surface (for further descriptions see text).**

| Sensor carrier | Sensor type | Water depth | Sensor unit manufacturer |
|---|---|---|---|
| FerryBox | Water temperature (°C) | 11 m | SBE45 |
| | Conductivity (ms m$^{-1}$) / Salinity (PSU)* | | SBE45 |
| | Oxygen (%) | | Anderra |
| | Chl-A (mg m$^3$) | | Cyclops |
| | pH | | Meinsberg |
| | Turbidity (FTU) | | Seapoint |
| Underwater node | Current (ADCP Teledyne Workhorse 600 kHz) | 13 m | Teledyne |
| Underwater node | Stereo-optical imaging system RemOs1 | Profiling[**] | Fischer et al., (2007) |
| Underwater Node | Pressure (dbar) | Profiling[**] | Sea&Sun CTD90 |
| | Water temperature (°C) | | |
| | Conductivity (ms m-1) / Salinity (PSU)* | | |
| | Oxygen (%) | | |
| | Chl-A (mg m$^3$) | | |
| | Turbidity (FTU) | | |

\* Calculated after actual UNESCO procedures.

\*\* between 11m water depth and surface.