# Peer review of "First year of the new Arctic AWIPEV-COSYNA cabled Underwater Observatory in Kongsfjorden, Spitsbergen."

_Ocean Science, 2016_

## Referee Comment (RC1) · C. Wiencke (Referee) · 3 Aug 2016

This paper reports about the first experiences obtained with the newly installed cabled underwater observatory installed in Kongsfjorden at the AWIPEV-base in Ny Ålesund, Spitsbergen. The system allows to measure temperature, salinity and turbidity and to make stereoscopic images at depths between 0 and 11 m water depth in the littoral of Kongsfjorden. The observatory works continuously and the image pairs and the hydrographic data are transferred automatically via Internet to the home laboratory in Germany for further processing. I can only congratulate the authors for the installation of this observatory in the Arctic allowing year-round observations of the shallow sub-littoral! The results obtained during the first year are breathtaking! For the first time

a distinct winter community was detected in the sahllow littoral, which by far exceeds the summer community in abundance and species diversity. The data show clearly that species abundance and species richness is highest during polar night between December and April. Beside the light conditions water temperature may influence the temporal occurrence of the winter community. The results obtained during the first year can give only a small glimpse of what can be expected in future especially with respect to the temporal and spatial dynamics of the higher trophic levels of the food web such as fish and macroinvertebrates. Therefore, I recommend to accept this manuscript as final paper in OS after minor revision. I recommend the following changes: Title: Change into "First year practical experiences…" Page 2, lines 24, 25: cite: Paar M, Voronkov A, Hop H, Brey T, Bartsch I, Schwanitz M, Wiencke C, Lebreton B, Asmus R, Asmus H (in press) Temporal shift in biomass and production of macrozoobenthos in the macroalgal belt at Hansneset, Kongsfjorden, after 15 years. Polar Biol. doi 10.1007/s00300-015-1760-6

Page 14, lines 110, 11: Cite: Müller R, Bartsch I, Laepple, T, Wiencke C (2011) Impact of oceanic warming on the distribution of seaweeds in polar and cold-temperate waters. In: Wiencke C (ed): Biology of Polar benthic algae, de Gruyter, p. 237-270

P14 line 23: This is not only the first data set from Kongsfjorden but for the whole Arctic. Do not hide your light under a bushel. This is a point you should stress!

There are numerous typing errors. Please change! The first letter in "Arctic" is always a capital letter! "Appendicularia" are spelled with a "double p"! Use either the term "Kongsfjord" or "Kongsfjorden"! The upperpanel of Fig. 5 is not necessary!

In the following I list the typos in correct spelling. Page 1, line 19: system Page 1, line 25: underwater Page 3, line 4: zooplankton Page 3, line 24: seasonally Page 3, line 25: Jørgensen and Gulliksen Page 4, line 7: remotely Page 4, line 18: trophic Page 5, line 18: Saccharina, algal, Phycodrys Page 8, line 2: cover Page 8, line 3: litho-pelagic Page 8, line 9: depth Page 8, line 11: allows Page 8, line 14, provides Page 10, line 10:

oculata Page 13, line24: insufficient Page 14, line 27: reveal Page 14 line 30: show Page 15, line 1: chaetognaths Page 15, line 11: sunlight Page 15, line 14: column Page 14, line 22: show Page 14, line 24: seaspider Page 15, line9: appendicularia page 17, line 31: shows page 18, line 18: loosing page 18, line 28: operation page 18, line 29: site page 21, line 20: Jørgensen page 25, line 27: reveal legend Fig. 8: seasonal

Further changes: Page 1, line 32: delete "the" before "polar systems" Page 5, line 17: change "Alaria esculenta" into italics Page 7, lines 26-28: Put "(Bartsch et al., 2016) at the end of the sentence. Page 9, line 3: Insert "also" after "shows". Page 9, line 11: Exchange "individuals" by "individual organisms" Page 13, line 1: Insert "fish" after "water". Page 13, line 20: Delete "the" after "that". Page 13, line 25: exchange "lead to" by "result in". Page 14, line 13: Exchange "did" by "do". Page 15, line 29: Exchange "this" by "these". Page 15, line 30: Change the order of words into: "may also". Page 16, line 21: Make a full stop after (Pihl, 1982). Delete "and assumed" by "This has been regarded as" Page 17, line 17: Change "Myoxocephalus scorpius" into italics. Page 17, line 30: Exchange "those" by "fish with these characteristics". Page 23, line 21: The paper appeared in "Polar Biol. 24, 2001, 113-121." Page 23, line 22: Change "Calanus" into italics. Page 23, line 13: The correct citation for this issue is: "Ber. Polarforsch. Meeresforsch. 492, 2004, 1-244"

---

## Referee Comment (RC2) · I. Puillat-Felix (Referee) · 16 Dec 2016

I fully agreed with the comments of the other referee. I would just add the following. This manuscript is valuable to be published because, in addition to its quality, it demonstrates how observatories are highly relevant to lead new researches when there are based on the deployment of several kinds of platforms together, with a multidisciplinary approach integrating biology with the understanding of the physical environment. It demonstrates how the technical challenge to deploy a cabled observatory with profiling systems including video profiler systems can be successfully addressed.

I consider our research and our society would benefit a lot of this type of research and that our responsibility is to promote it and to express a strategy for its future use. Consequently i would recommend to add a specific paragraph at the end of the conclusion: next steps and needs.

---

## Author Response (AR1)

**Dear editor,**

please find attached the revised version of the ms according to the recommendations of the two reviewers. We were able to fulfill all reviewers' recommendations and comments and have made all changes available in the ms in a tracking version. Attached to this letter and according to the instructions, I have copied the revised version with tracking option active at the end of this letter. A clean version is sent to you in the manuscript option in your homepage. I hope I understood the procedure correctly. If not, I can immediately also send the doc files.

The following changes according to the reviewers' recommendations were made in the revised version:

**Comments of reviewer 1 and the authors responses:**

I recommend to accept this manuscript as final paper in OS after minor revision. I recommend the following changes: Title: Change into "First year practical experiences. . ." **Authors comment: Changed in the revised version**

Page 2, lines 24, 25: cite: Paar M, Voronkov A, Hop H, Brey T, Bartsch I, Schwanitz M, Wiencke C, Lebreton B, Asmus R, Asmus H (in press) Temporal shift in biomass and production of macrozoobenthos in the macroalgal belt at Hansneset, Kongsfjorden, after 15 years. Polar Biol. doi 10.1007/s00300-015-1760-6 Authors comment: cited in the revised version

Page 14, lines 110, 11: Cite: MuÌ Ller R, Bartsch I, Laepple, T, Wiencke C (2011) Impact of oceanic warming on the distribution of seaweeds in polar and cold-temperate waters. In: Wiencke C (ed): Biology of Polar benthic algae, de Gruyter, p. 237-270 **Authors comment: cited in the revised version**

P14 line 23: This is not only the first data set from Kongsfjorden but for the whole Arctic. Do not hide your light under a bushel. This is a point you should stress! **Authors comment: changed in the revised version**

**Authors comment: All following typos and spelling errors have been corrected in the revised version:**

There are numerous typing errors. Please change! The first letter in "Arctic" is always a capital letter! "Appendicularia" are spelled with a "double p"! Use either the term "Kongsfjord" or "Kongsfjorden"! The upperpanel of Fig. 5 is not necessary! In the following I list the typos in correct spelling. Page 1, line 19: system Page 1, line 25: underwater Page 3, line 4: zooplankton Page 3, line 24: seasonally Page 3, line 25: Jørgensen and Gulliksen Page 4, line 7: remotely Page 4, line 18: trophic Page 5, line 18: Saccharina, algal, Phycodrys Page 8, line 2: cover Page 8, line 3: litho-pelagic Page 8, line 9: depth Page 8, line 11: allows Page 8, line 14, provides Page 10, line 10: oculata Page 13, line24: insufficient Page 14, line 27: reveal Page 14 line 30: show Page 15, line 1: chaetognaths Page 15, line 11: sunlight Page 15, line 14: column Page 14, line 22: show Page 14, line 24: seaspider Page 15, line9: appendicularia page 17, line 31: shows page 18, line 18: loosing page 18, line 28: operation page 18, line 29: site page 21, line 20: Jørgensen page 25, line 27: reveal legend Fig. 8: seasonal Further changes: Page 1, line 32: delete "the" before "polar systems" Page 5, line 17: change "Alaria esculenta" into italics Page 7, lines 26-28: Put "(Bartsch et al., 2016) at the end of the sentence. Page 9, line 3: Insert "also" after "shows". Page 9, line 11: Exchange "individuals" by "individual organisms" Page 13, line 1: Insert "fish" after "water". Page 13, line 20: Delete "the" after "that". Page 13, line 25: exchange "lead to" by "result in". Page 14, line 13: Exchange "did" by "do". Page 15, line

29: Exchange "this" by "these". Page 15, line 30: Change the order of words into: "may also". Page 16, line 21: Make a full stop after (Pihl, 1982). Delete "and assumed" by "This has been regarded as" Page 17, line 17: Change "Myoxocephalus scorpius" into italics. Page 17, line 30: Exchange "those" by "fish with these characteristics". Page 23, line 21: The paper appeared in "Polar Biol. 24, 2001, 113-121." Page 23, line 22: Change "Calanus" into italics. Page 23, line 13: The correct citation for this issue is: "Ber. Polarforsch. Meeresforsch. 492, 2004, 1-244"

**Comments of reviewer 2 and the authors responses:**

I fully agreed with the comments of the other referee. I would just add the following. This manuscript is valuable to be published because, in addition to its quality, it demonstrates how observatories are highly relevant to lead new researches when there are based on the deployment of several kinds of platforms together, with a multidisciplinary approach integrating biology with the understanding of the physical environment. It demonstrates how the technical challenge to deploy a cabled observatory with profiling systems including video profiler systems can be successfully addressed. I consider our research and our society would benefit a lot of this type of research and that our responsibility is to promote it and to express a strategy for its future use. Consequently, I would recommend to add a specific paragraph at the end of the conclusion: next steps and needs.

**Authors comment: We have added a "Next steps and needs" paragraph to the end of the discussion section and tried to include all recommendations from reviewer 2 in this passage.**

**New paragraph on page 18 - 19:**

Next steps and needs Besides the here presented ecological and hydrographical results from the Kongsfjorden ecosystem, the study demonstrates the advantages of permanently operated cabled observatory technology - especially when combined with other research methods in a multidisciplinary approach integrating biology with the understanding of the physical environment. Cabled observatories with continuous power supply and network access allow the use of state of the art IT-technology and smart-monitoring approaches under water. These are often not applicable in mooring based sensor technology because no feedback to the operator is possible and thereby the researcher itself cannot react on specific environmental situations during the measuring process. Furthermore, complex sensor systems like profiling video or stereo-imaging systems often cannot be operated unsupervised for longer times because the controlling software is either too complex, the power consumption is too high, or the required test and development phases for an unsupervised operation of such systems are too long and therefore too expensive. Cabled observatories with permanent access, power supply and systems control allow even complex sensor systems to be operated for longer periods because in case failures, the system can give an alert to an operator elsewhere to request remote control and if necessary sensor reset. Based on our experiences with the cabled observatory in Svalbard, we assume that such underwater research facilities, if operated within an international and well-focused research strategy, may significantly promote our knowledge especially in remote and sensitive areas like the polar regions.

[revised manuscript text omitted]

salinity, pH, Chl-A and turbidity and additionally made a quantitative analysis of the abundance, species occurrence and (for selected species) length-frequency distribution of the fish and macroinvertebrate taxa. For the latter assessment, a stereo-optical macro-biota observatory called "RemOS1" (Remote Optical System) was used

- 5 specifically designed for long-term exposure and assessments of fish and macroinvertebrate communities in shallow water areas (Fischer et al., 2007b). Data acquisition was conducted year round, remote controlled with a temporal resolution of 1Hz for the hydrological data and a stereoscopic imaging frequency of 30 min. Parallel to this study, classic fishing campaigns were performed in the same area with
- 10 standard fyke-nets to provide ground-truth data for the remotely sampled fish data. The fishing campaigns were performed in June/July and September of 2012 to 2014. The results of 2012 to 2013 are published in Brand & Fischer (2016). The publication of the dataset of 2014 together with an comparative analysis of the results of the UWO is in preparation (Brand, pers. comm.)
- 15

The present study aims to demonstrate the high potential of remote controlled sensors to quantitatively assess not only hydrological data such as temperature, current or plankton community with classical CTD probes or VPR recorders but also for the assessment of higher tropic levels such as macro-invertebrates and fish. To the best of

- 20 our knowledge, there are only a small number of studies and observatories available world-wide that are trying to assess also higher trophic levels with remote controlled optical systems (Aguzzi et al., 2011; Buckland et al., 2005; Fischer et al., 2007b; Wehkamp and Fischer, 2014) and even fewer in regard to quantitative assessments with respect to specimen's abundances and species specific length-frequency analysis
- 25 in an area. Because these technologies will certainly develop and improve over the next years, this study also discusses certain specific requirements and challenges for such systems, especially for shallow water artic areas.

**2 Materials and Methods**

30 The UWO was built up in 2012 in the framework of COSYNA (Coastal Observing Systems of the Northern and Arctic Seas). The system comprises a land based FerryBox system equipped with various hydrographic sensors (Table 1) receiving water from a remote controlled underwater pump station at 11 m water depth. Additionally, a cable connected (fibre-optic and 240V power) underwater node (Fig.

[revised manuscript text omitted]

during a restricted period of time in the polar summer when light is available and the temperatures allow for *in situ* methods. Therefore, our knowledge on polar shallow water ecosystems and especially their role as nursery and juvenile habitat is extremely restricted. Most of the recent studies (e.g., Hop *et al.* 2002; Svendsen *et al.* 2002; Hop

- 5 et al. 2006) in our addressed study area have been conducted during summer, when the ford system is accessible by research vessels. Although the summer productive period is of great importance for Arctic ecosystems, several crucial processes (e.g., reproduction) take place during other seasons and especially during the polar winter. During these times, however, almost no information is available in most Arctic fjord
- 10 systems (Kwasniewski, 2003). Understanding polar ecosystems in the context of global warming and expected or already observed ecosystem changes (Müller et al., 2011, Bartsch et al., 2016) is, however, crucial for thoroughly understanding the ecosystem behaviour in polar areas.
- In this study, we do not provide results from experimental work in Kongsfjorden
  based on discrete studies with a clear short term ecological hypothesis. In contrast, we provide data from a one year long quantitative assessment of hydrographic parameters together with quantitative data on the macrobiota community assessed by a remote controlled cable-connected underwater observatory installed in a typical shallow water habitat in the Kongsfjorden. Using a remote controlled vertical profiling
- 20 system, we were able to continuously assess temperature, salinity, turbidity and other hydrographic parameters together with the shallow water macrobiotic community over the entire water column from the benthic over the epi-benthic to the pelagic realm in a high temporal resolution. To our knowledge, this is the first dataset both from Kongsfjorden but also from the entire Arctic that reveals such a year round
- 25 assessment of the shallow water macrobiotic community together with the quantitative data of the water temperature, salinity and turbidity and, therefore, allows a deeper insight in the coupling of the seasonal dynamics of the biology and the hydrography compared to pure summer studies. The data reveal a distinct winter community in the fjords shallow water ecosystem, which by far exceeds the summer
- 30 community in both, abundance and species diversity. Although we have not yet calculated biomass per m3 for the assessed species, our data clearly show, that the species abundance and species richness is highest during the polar winter that begins in December when no more light is available under water. During this time, except for

|---|----------------------------|

| Gel | ösch | t: dic |
|-----|------|--------|
|     |      |        |

|----------------|--|--|
|                |  |  |
|                |  |  |
|                |  |  |
|                |  |  |
|                |  |  |
|                |  |  |
|                |  |  |
|                |  |  |
|                |  |  |

the appendicularia, most species including fish (mainly gadids of the species *Gadus morhua* and Boerogadus saida), jellyfish (mainly *Beroe sp.*), chaetognaths (*Parasagitta elegans*), pteropods (*Clione limacina*) and smaller benthic and epibenthic crustacean (most possibly *Mysis oculata*, C. Buchholz pers. comm.) invade

- 5 the shallow water zone and build up highest abundances. During this study, an overall peak abundance was observed in February when the common sea spider *Hyas araneus* clearly dominated the community in numbers and biomass for a short time. Only one month later in March, however, *Hyas araneus* almost completely disappeared when fish, jellyfish and pteropods formed the predominant community with respect to the
- 10 overall abundances. The "winter" community persisted until April and then almost vanished. The time of the winter community "disappearance" highly corresponds with the increasing availability of light under water. Although sunlight is available at NyAlesund again already during mid of March (http://www.awipev.eu/awipevobservatories/current-weather/), the inclination angle of the light is still low until
- 15 April so that only a small fraction of the sunlight is penetrating the water column, (personal observation). However, to really correlate the presence of the "winter community" with the availability of light underwater, discrete measurements of the light intensity and light quality are necessary in the different depth strata to reveal if light is an ultimate factor for the temporal occurrence of the fjords shallow water
- 20 winter community or only a proxy associated with other environmental factory. Our data suggest that especially water temperature may also have a significant influence on the spatio-temporal occurrence of the winter community. Our daily sampled temperature profiles clearly show, that water temperature in the shallow water areas of Kongsfjorden can change within short times, even in winter, between < 0°C and up to
- 25 4°C. In particular, the peak abundance in the common seaspider *Hyas araneus* corresponds with the time of higher water temperature during February, and the collapse of the spider abundance occurred when the water temperatures decreased from 4°C to only approximately 2°C again. Similar temporal pattern could be observed also in the overall species abundance in April when a short cold phase in the
- 30 water temperature occurred. However, these seemingly corresponding changes in the biotic community and the changes in the abiotic environments may also be purely by chance and we do not know yet if there are functional relationships between these observations. The permanent installation of the cabled underwater observatory at NyAlesund allows to formulate and test such a hypothesis of a persisting shallow

|----------------------------------------------------------------------------------------|
%C3%85lesund_observation_site/ |
|                                                                                        |

|----|------------------|
|    |                  |
| _  |                  |
|    |                  |
|    |                  |
|    |                  |
|    |                  |
|    |                  |
|    |                  |
| C  | 0-14             |
| Ľ  |                  |

water "winter community" in the fjord system as well as the hypothesized controlling or at least affecting abiotic factors.

Our data additionally reveal another distinct community during the summer months when the temperatures increased up to 8°C in the fjord. Then, appendicularia occurred in higher abundances for a restricted time, i.e., from August to October, in the shallow water with a peak in abundances in September. In contrast to the winter community, which was mainly benthic or at least benthos-associated, this summer community was almost completely dominated by a single appendicularia species, most certainly

belonging to the genus *Oikopleura sp.* (Dahms et al., 2015).
Besides appendiculria, also juvenile cod fish were found in September in the deeper littoral water layers closely associated to benthic habitats. The detailed length

5

- frequency analysis of this cohort reveal, that these fish were the YOY offspring of the same year (YOY cohort 2014) with an average standard length of  $65 \pm 16$  mm. The data also reveal, that these fish seem to stay in the littoral zone (even though the
- 15 overall abundances strongly decreased over winter) and continuously grow and reach an average standard length of 100 to 125 mm in February – March at age-class 1 when they seem to quantitatively leave the shallow water habitats. This outcome indicates a complex migration pattern of YOY cod in this area with a short winter phase in the littoral zone of the fjord system of Spitzbergen and a later migration
- 20 towards deeper or offshore habitats as adults. Such temporally restricted shallow water phases have been observed already for several other cod species, especially during their juvenile phase (Pihl, 1982). This has been regarded as a juvenile behaviour to prevent predation by older conspecifics in the deeper adult habitats (Ruiz et al., 1993) as well as an improvement in foraging efficiency of the juveniles during
- 25 their non-piscivore microzoobenthic benthic feeding phase (Pihl, 1982).

In contrast to the clearly visible seasonal growth pattern in the cod species, no distinct growth could not be observed in none of the other species even in the highly abundant common sea spider, which showed a persisting size range between approximately 50 and 80 mm during the entire winter month except for the month of November in both

30 years when larger animals between 120 and 180 m were observed in the area, even though in much lower abundances.

As clearly stated before, this study does not provide a singular hypothesis driven

| Formatiert: | Englisch | (USA) |
|-------------|----------|-------|
|-------------|----------|-------|

|----------------------------------------------|

question; instead, it focuses on a basic assessment of the temporal (and with respect to the water column also spatial) pattern in the macrobiota community distribution and possible hydrographic factors that influence the shallow water biota. The results of this study are by far incomplete and only represent a one-year study at a specific site

- 5 in the Kongsfjorden ecosystem, which may or may not be representative of the shallow water community of this area. However, the study presents a continuous year round data set in a temporal resolution of one week, which is, to our knowledge, not available in any other fjord system and especially not in the Arctic environment where winter data are missing at almost every level. However, even though the data provide
- a unique year round insight in a polar shallow water fjord community, we can assume that the technology used here has a certain bias with respect to species selectivity. Therefore, these data have to be taken with care. For instance, comparing our stereo-optically assessed fish data with data from classical sampling devices in Kongsfjord (Brand and Fischer, 2016; Hop et al., 2002; Renaud et al., 2011) or even with sporadic
- 15 divers observations (Brand and Fischer, 2016; Hop et al., 2002), it becomes clear that also our optical sensors are species selective. Brand and Fischer (2016), for example reported for the summer month a distinct occurrence of the benthic sculpin *Myoxocephalus scorpius*, a typical temperate and highly camouflaged benthic fish species in fyke-net catches. Although we detected *Myoxocephalus scorpius* during
- 20 summer also on the stereoscopic images, the overall abundance remained quite low. Unfortunately, the fyke net catches of Brand and Fischer (2016), as most other available marine studies of the fjord, are only available for the polar summer months when our stereo-optical data revealed the lowest overall biota abundance at all. However, taking into account that fyke-nets are highly time integrative and catch fish
- 25 only directly at the bottom, the fyke-net and optical data may be rather complementary than contradictory. In the study of Brand and Fischer (2016), fyke nets with a mesh size of 12 mm and a steering net of 18 mm were used. This type of net gear is highly selective for strictly benthic fish species with a high potential of entanglement, such as sculpins. In contrast, a stereo-optical method is most probably
- 30 less selective for benthic highly camouflaged fish species and may significantly underestimate fish with these characteristics,

Instead, our overall image assessment procedure was thoroughly performed by two different persons and showed similar results with respect to the quantitative detection Gelöscht: a

|-------------------|--|

of even small benthic mysids. Therefore, we assume that we would have detected also sculpins if available in higher abundances and thus conclude that the quantitative relation of the average abundance between the major fish species found on the images might be more precise as found in the fyke net catches. This outcome seems to be

- 5 supported also by the available diver observations in that area at least during summer. Hop et al. (2002) and Renaud et al. (2011) both reported the cod species *Gadus morhua* as one of the most abundant species in the area which would be in accordance with our findings. Nevertheless, the comparison of these two methods shows that there is a large uncertainty with respect to the methodological approach that should be
- 10 used in future studies. Furthermore, our *in situ* optical methods allow for a lowinvasive abundance estimate, for a precise length-frequency analysis of the mapped fish and also for a continuous year round assessment of the community. However, it does not allow for further investigations such as stomach content analysis and precise aging based on scale or otolith analysis. Jf we manage to combine such continuous
- 15 hydrographic and community observations using cable-connected observatories with classical ground trouthing fishing or sampling methods, we may reduce our scientific fishing effort to a limited number of specimen, which are needed for specific detailed analysis such as stomach content and otolith-based aging and obtain the required more invasive stock abundance and growth data via non-invasive optical methods. These
- approaches may finally enable the reduction of our fishing effort without loosing the required data density and therefore contribute to the increasing scientific demand of a resource conservative science also in fish and community ecology, especially in ecological sensitive areas, such as the polar fjords or marine protected areas.

**Next steps and needs**

- 25 Besides the here presented ecological and hydrographical results from the Kongsfjorden ecosystem, the study demonstrates the advantages of cabled observatory technology - especially when combined with other research methods in a multidisciplinary approach integrating biology with the understanding of the physical environment. Cabled observatories with continuous power supply and network access
- 30 allow the use of state of the art IT-technology and smart-monitoring approaches under water. These are often not applicable in mooring based sensor technology because no feedback to the operator is possible and thereby the researcher itself cannot react on specific environmental situations during the measuring process. Furthermore, complex sensor systems like profiling video or stereo-imaging systems often cannot

|---|------------------------------------------------------------------------------------------------------|

be operated unsupervised for longer times because the controlling software is either too complex, the power consumption is too high, or the required test and development phases for an unsupervised operation of such systems are too long and therefore too expensive. Cabled observatories with permanent access, power supply and systems

5 control allow even complex sensor systems to be operated for longer periods because in case failures, the system can give an alert to an operator elsewhere to request remote control and if necessary sensor reset. Based on our experiences with the cabled observatory in Svalbard, we assume that such underwater research facilities, if operated within an international and well focused research strategy, may significantly

10 promote our knowledge especially in remote and sensitive areas like the polar regions.

|------------------------|------------------------------|
|                        |                              |

**Competing interests**

15

20 The authors declare that they have no conflict of interest.

**Acknowledgments**

[revised manuscript text omitted]

25 Müller, R., Bartsch, L. Laepple, T., Wiencke, C., Impact of oceanic warming on the distribution of seaweeds in polar and cold-temperate waters. In: Wiencke C (ed): Biology of Polar benthic algae, de Gruyter, p. 237-270, 2011,

Paar, M., Voronkov, A., Hop, H., Brey, T., Bartsch, I., Schwanitz, M., Wiencke, C., Lebreton, B., Asmus, R. and Asmus, H.: Temporal shift in biomass and production of

in arctic Kongsfjord (Svalbard) studied by means of s
and photography, Polar Biol, 2001.                                                                                                                                                                                                                                       | cy bottom fauna suction sampling                                                                         |
|-----------------------------------------------------------------------------------------------------------------------------------------------------------------------------------------------------------------------------------------------------------------------------------------------------------------------------------------------------------------------------------------|----------------------------------------------------------------------------------------------------------|
Englisch (USA)                                                                                                                                                                                                                                                                                                                         | Roman,                                                                                                   |
Schriftfarbe: Automatisch                                                                                                                                                                                                                                                                                                              | Roman, 12 pt,                                                                                            |
Schriftfarbe: Automatisch                                                                                                                                                                                                                                                                                                              | Roman, 12 pt,                                                                                            |
Schriftfarbe: Automatisch                                                                                                                                                                                                                                                                                                              | Roman, 12 pt,                                                                                            |
Schriftfarbe: Automatisch                                                                                                                                                                                                                                                                                                              | Roman, 12 pt,                                                                                            |
Schriftfarbe: Automatisch                                                                                                                                                                                                                                                                                                              | Roman, 12 pt,                                                                                            |
Schriftfarbe: Automatisch                                                                                                                                                                                                                                                                                                              | Roman, 12 pt,                                                                                            |
Schriftfarbe: Automatisch                                                                                                                                                                                                                                                                                                              | Roman, 12 pt,                                                                                            |
Schriftfarbe: Automatisch                                                                                                                                                                                                                                                                                                              | Roman, 12 pt,                                                                                            |
Schriftfarbe: Automatisch                                                                                                                                                                                                                                                                                                              | Roman, 12 pt,                                                                                            |
Schriftfarbe: Automatisch, Englisch (USA)                                                                                                                                                                                                                                                                                              | Roman, 12 pt,                                                                                            |
Schriftfarbe: Automatisch, Englisch (USA)                                                                                                                                                                                                                                                                                              | Roman, 12 pt,                                                                                            |
Englisch (USA)                                                                                                                                                                                                                                                                                                                         | Roman,                                                                                                   |
1.5 Zeilen, Keine Absatzkontrolle, Leerraum zw
asiatischem und westlichem Text nicht anpasseu
zwischen asiatischem Text und Zahlen nicht anp
Tabstopps: 0,99 cm, Links + 1,98 cm, Links +
+ 3,95 cm, Links + 4,94 cm, Links + 5,93 cm
cm, Links + 7,9 cm, Links + 8,89 cm, Links +
+ 10,86 cm, Links + 11,85 cm, Links | Zeilenabstand:
ischen
n, Leerraum
passen,
2,96 cm, Links
, Links + 6,91
9,88 cm, Links |

macrozoobenthos in the macroalgal belt at Hansneset, Kongsfjorden, after 15 years, Polar Biol, 39(11), 2065–2076, doi:10.1007/s00300-015-1760-6, 2015.

[revised manuscript text omitted]